# Effect of Dexamethasone on the Incidence and Outcome of COVID-19 Associated Pulmonary Aspergillosis (CAPA) in Critically Ill Patients during First- and Second Pandemic Wave—A Single Center Experience

**DOI:** 10.3390/diagnostics12123049

**Published:** 2022-12-05

**Authors:** Simon Dubler, Ömer Can Turan, Karsten Daniel Schmidt, Peter-michael rath, Hedda-Luise Verhasselt, Sandra Maier, Annabell Skarabis, Thorsten Brenner, Frank Herbstreit

**Affiliations:** 1Department of Anesthesiology and Intensive Care Medicine, University Hospital Essen, University Duisburg-Essen, Hufelandstrasse 55, 45147 Essen, Germany; 2Institute of Medical Microbiology, Essen Centre of Excellence in Clinical and Laboratory Mycology and Clinical Studies, University Hospital Essen, University Duisburg-Essen, Hufelandstrasse 55, 45147 Essen, Germany; 3Institute of Diagnostic and Interventional Radiology and Neuroradiology, University Hospital Essen, University Duisburg-Essen, Hufelandstrasse 55, 45147 Essen, Germany

**Keywords:** COVID-19, CAPA, *Aspergillus fumigatus*, dexamethasone, intensive care medicine, ICU

## Abstract

Superinfections with *Aspergillus* spp. in patients with Coronavirus disease 2019 (CAPA: COVID-19-associated pulmonary aspergillosis) are increasing. Dexamethasone has shown beneficial effects in critically ill COVID-19 patients. Whether dexamethasone increases the risk of CAPA has not been studied exclusively. Moreover, this retrospective study aimed to identify risk factors for a worse outcome in critically ill COVID-19 patients. Data from 231 critically ill COVID-19 patients with or without dexamethasone treatment from March 2020 and March 2021 were retrospectively analysed. Only 4/169 (6.5%) in the DEXA-group and 13/62 (7.7%) in the Non-DEXA group were diagnosed with probable CAPA (*p* = 0.749). Accordingly, dexamethasone was not identified as a risk factor for CAPA. Moreover, CAPA was not identified as an independent risk factor for death in multivariable analysis (*p* = 0.361). In contrast, elevated disease severity (as assessed by Sequential Organ Failure Assessment [SOFA]-score) and the need for organ support (kidney replacement therapy and extracorporeal membrane oxygenation [ECMO]) were significantly associated with a worse outcome. Therefore, COVID-19 treatment with dexamethasone did not increase the risk for CAPA. Moreover, adequately treated CAPA did not represent an independent risk factor for mortality. Accordingly, CAPA might reflect patients’ severe disease state instead of directly influencing outcome.

## 1. Introduction

In 2019, severe acute respiratory syndrome coronavirus 2 (SARS-CoV-2) emerged in Wuhan, China [1] and became a pandemic. Around 1% of patients infected with SARS-CoV-2 [Coronavirus disease 2019 (COVID-19)] needed intensive care treatment [2]. In a large retrospective analysis from Germany, proportion of patients admitted with COVID-19 and requiring ICU treatment dropped from 30% during the first wave to 14% during the second and third waves, respectively [3].

Patients with severe courses of viral infections like influenza are at increased risk of secondary bacterial and fungal infections, including invasive pulmonary aspergillosis (IPA) caused by *Aspergillus* spp. [4]. In a retrospective study by Schauwvlieghe et al. [5] IPA was diagnosed in 14% of non-immunocompromised patients with influenza, while only 5% in the control group developed IPA. Mortality was as high as 51% in influenza patients with IPA. Influenza was also independently associated with IPA, along with the use of corticosteroids, before admission to the intensive care unit (ICU). 

Critically ill patients with COVID-19 seem to be another group at risk for IPA. Increasing reports about COVID-19-associated pulmonary aspergillosis (CAPA) during the pandemic raised concerns due to it being a contributing factor for mortality [6]. Depending on the diagnostic workup and the applied definition, incidence of CAPA varied between 10% and 26% [7,8,9,10] with mortality rates ranging from 44% to 71% [7,8,9,10,11,12,13]. In a multicentre study using worldwide data (Fungiscope) during the first wave of the pandemic, CAPA incidence ranged from 1.0% to 39.1%. Mortality was 52.2%, whereas 33% of these deaths were attributed to CAPA [14].

The severity of COVID-19 depends on many physiological factors of which some are explainable in artificial intelligence (AI)-assisted severity classification. In a study by Liu et al. [15], 78 patients with COVID-19 were categorized as mild, regular, severe, and critical according to clinical conditions. Early triage of these patients could be supported by machine learning with CT-based clinical-radiological features. Timely diagnosis of severe and critically ill cases is essential to ensure early intervention and [14] therefore reduce the mortality of these patients

Numerous pharmacological treatments failed to show beneficial effects on survival in patients with severe COVID-19. The greatest treatment success in this patient group was achieved by the early administration of dexamethasone in the RECOVERY trial with over 6000 patients [16]; patients in the dexamethasone group (6 mg dexamethasone daily for ten days) showed a lower incidence of death within 28 days than the usual care group (22.9% versus 25.7%, *p* < 0.001). The effect was most pronounced in critically ill patients with invasive ventilation or oxygen therapy with reductions in mortality of 12% and 3%, respectively. A meta-analysis including several randomized controlled trials (RCT) including the RECOVERY trial confirmed this effect [17].

Dexamethasone is a potent glucocorticoid with immunomodulatory properties that impair innate immune functions by a reduction in the production of reactive oxidant species (ROS) and LC3-associated phagocytosis (LAP) [18]. This might lead to an increased susceptibility for invasive infections caused by *Aspergillus* spp. An effect on the incidence of CAPA was not studied in the RECOVERY trial.

There is conflicting data on whether the use of corticosteroids increases the risk of CAPA in patients with severe COVID-19. In two large series of CAPA patients by White et al. [7] and Bartoletti et al. [19], the use of high doses of corticosteroids increased the likelihood of CAPA. Other studies [11,20,21,22] could not show any negative influence of corticosteroid use in patients with CAPA.

We therefore conducted a single centre trial examining the association of dexamethasone and the incidence and outcome of CAPA in a relatively large cohort of critically ill patients with COVID-19 in Germany.

## 2. Materials and Methods

This study was conducted as a monocentric, retrospective study at the ICU of the department of anaesthesiology and intensive care medicine, University Hospital Essen (Essen, Germany). It was approved by the local ethics committee of the Medical Faculty of the University of Essen-Duisburg (20-9368-BO) and was registered at the German clinical trials register (DRKS-ID: DRKSS00026071).

Adult patients admitted to the ICU between March 2020 and March 2021 (during the first and second pandemic waves) with confirmed SARS-CoV-2 infection (COVID-19) and need for any respiratory support (minimum of 2 litres O_2_ via nasal cannula) were included in the study. Patients who died within six days of ICU admission were excluded. SARS-CoV-2 was detected by point-of-care polymerase chain reaction (PCR) testing (Cepheid GeneXpert, Cepheid GmbH, Krefeld, Germany). Sampling was performed as bronchoalveolar lavage (BALF) in intubated patients. Non-intubated patients received nasopharyngeal swabs. Baseline characteristics were compared between patients with and without dexamethasone treatment. The main clinical endpoint was the incidence of COVID-19-associated pulmonary aspergillosis (CAPA) according to the latest definitions by the European Excellence Center for Medical Mycology (ECMM) and the International Society for Human and Animal Mycology (ISHAM) consensus criteria [6]. Secondary endpoints were Co-morbidities [Charlson-Comorbidity Index (CCI)], mortality, length of ICU and hospital stay, need for invasive ventilation, and extracorporeal membrane oxygenation (ECMO) therapy. To quantify the severity of disease, we used the new Simplified Acute Physiology Score 2 (SAPS II), as well as the Sequential Organ Failure Assessment (SOFA) Score at ICU admission.

### 2.1. Treatment protocols

All patients admitted to the ICU with COVID-19 were treated following a standardized treatment protocol as published previously [23]. Since July 2020, all patients at our institution received 6 mg dexamethasone intravenously (IV) or per oral route (PO) for at least six days (and a maximum of ten days). This remains the latest and actual standard of care at our institution. Other specific treatments are listed in Appendix A. 

### 2.2. Definitions

#### 2.2.1. CAPA

The 2020 ECCM/ISHAM consensus criteria were used for *proven*, *probable*, or *possible* CAPA diagnosis [6]. Since lung biopsies were not performed in this patient population, there was no *proven* CAPA case in our cohort. Since all patients with invasive mechanical ventilation received bronchoalveolar lavage (BALF) for diagnosis of CAPA, categorization into *possible* CAPA (this definition depends on non-bronchoscopy lavage) was not suitable. *Probable* CAPA (pCAPA*) according to the consensus criteria was considered as a diagnosis of CAPA in this study. Patients with direct or indirect evidence of *Aspergillus* spp. but not fulfilling ECCM/ISHAM criteria were declared “colonization” (=Non-CAPA). A radiologist (S.M.) examined chest computed tomography (CT) scans of all patients with direct or indirect evidence of invasive fungal infections and checked these scans for signs of IPA, according to the diagnostic algorithm by ECMM/ISHAM consensus criteria [6].

* pCAPA Diagnosis: Clinical factors (refractory fever or pleural rub or chest pain or hemoptysis or a combination) + Imaging (pulmonary infiltrates or cavitating infiltrate or both) + microbiology [*Aspergillus* spp. by microscopy in BALF or *Aspergillus* spp. culture in BALF or GM > 0.5 ODI in Serum or GM > 1.0 ODI in BALF or 2× *Aspergillus* spp. by PCR in serum/plasma/whole blood or *Aspergillus* spp. by PCR in BALF (<36 threshold cycle) or *Aspergillus* spp. by PCR in serum/plasma/whole blood + *Aspergillus* spp. by PCR in BALF].

#### 2.2.2. Waves

Classification of different waves during the COVID-19 pandemic were determined according to local authorities [24] (Wave 1: March 2020–27 September 2020; Wave 2: 28 September–end of March 2021).

#### 2.2.3. ARDS

Acute respiratory syndrome (ARDS) was defined according to the latest Berlin definitions [25].

### 2.3. Sequential Organ Failure Assessment without Glasgow Coma Scale (SOFA Non-GCS)

The SOFA score describes the acute state of illness and predicts mortality in ICU patients based on the function of different organ systems [26]. Since assessment and documentation of the Glasgow Coma Score (GCS) in ICU patients is frequently uncertain and/or incomplete, we defined a modified non-GCS SOFA score for our data analyses, excluding the item GCS [27].

### 2.4. Outcome

“Survivors” were defined as patients discharged alive from the Intensive Care Unit (ICU).

### 2.5. Mycological Studies

According to standardized local protocols, all invasive ventilated patients received a CT scan of chest and abdomen following BALF sampling for microbiological testing at ICU admission in case of ARDS:▪Blood cultures▪SerumoGalactomannan (GM): twice weekly▪BALFoGram stain and immediate microscopyoCulturesoPCR (multiplex for common causes of pneumonia)oPCR (*Mycobacterium tuberculosis*, *Pneumocystis jirovecii*, *Legionella pneumophila*)oPCR [*Aspergillus* spp. and azole-resistance markers TR34 and TR46.: *Aspergillus fumigatus* (*A. fumigatus*), *A. terreus*, *A. flavus*]oPCR (viral panel, including Cytomegalovirus (CMV), Herpes-simplex virus [HSV-1 and HSV-2], Influenza-A, Influenza-B, Respiratory syncytial virus [RSV] and SARS-CoV-2) oGalactomannan (GM)▪UrineoCultureoAntigen testing for *L. pneumophila* and *Streptococcus pneumoniae*▪Nasal and rectal swaboScreening for Methicillin-resistant Staphylococcus aureus (MRSA), Vancomycin-resistant Enterococcus faecalis/faecium (VRE) and resistant gram-negative pathogens

GM testing was performed with Platelia™ *Aspergillus* ELISA (Bio-Rad, Marnes-la-Coquette, France) interpreted as a positive result at an optical density index (ODI) of 1.0 in BAL according to the manufacturer [28]. *Aspergillus* real-time PCR were tested with AsperGenius^®^ (PathoNostics, Maastricht, The Nederlands). Characterization, susceptibility testing, and interpretation of minimal inhibitory concentrations (MICs) of all *Aspergillus fumigatus* isolates was performed as described before by Kirchhoff et al. [29]. No Azole-resistant *Aspergillus* strain was detected in this study. Cultured isolates in this study only revealed *A. fumigatus*.

### 2.6. Pharmacological Studies and Therapeutic Drug Monitoring (TDM)

All patients receiving voriconazole as a first line anti-fungal therapy were screened for cytochrom-P-450 subclass (CYP2C19)-activity, according to the guidelines of the clinical pharmacogenetics implementation consortium (CPIC) [30]. In cases of rapid- or ultra-rapid metabolism, voriconazole was changed immediately to liposomal Amphotericin B. Therapeutic drug monitoring (TDM) (In-house BioAssay [31]) was performed for every patient with voriconazole therapy after a five-day treatment course. In cases of low therapeutic drug levels (targeted therapeutic drug levels for voriconazole were 2–6 µg/mL), the daily dose was increased and further TDM controls were conducted. 

## 3. Statistics

Baseline characteristics of the two groups were compared using standardized differences (SD). For the statistical analysis, Stata 16 (StataCorp LLC, College Station, TX, USA) was used. Continuous data were tested for normal distribution using Shapiro-Wilks test. Normally distributed data were analysed using the Student *t*-test. A Mann-Whitney-U-test was employed for not normally distributed data. Categorial results were analysed with a Chi-Square-test. Due to the retrospective character of the study, all analyses had exploratory intentions and *p* < 0.05 was considered significant.

A *t*-test was used with a *p* < 0.05 suggesting significant differences. For categorical, symmetrically distributed continuous, and non-normal continuous variables with Chi-quadrat, Student *t*-tests and Mann-Whitney tests were used to analyse further differences. 

## 4. Results

A total of 794 patients with confirmed infection with SARS-CoV-2 were treated at the University Hospital Essen (Germany) between March 2020 and March 2021. Of these 794 patients, 293 were treated in a specialized ICU facility capable of ECMO therapy in cases of severe ARDS. A total of 231 patients were included in the final analysis. The main reason for exclusion was missing respiratory support (21 cases) and treatment in multiple ICUs (12 cases). See Figure 1 for the study flow chart and patient inclusion and exclusion criteria. 

A total of 169 patients (73.2%) received 6 mg dexamethasone daily for at least six days (DEXA-Group), and 62 patients (26.8%) did not receive dexamethasone (non-DEXA group). Baseline characteristics of included patients are presented in Table 1. Age was not significantly different between the groups (*p* = 0.716). Patients were predominantly male in both groups with a higher proportion of female patients in the DEXA-Group (30.8% versus 14.5%, *p* = 0.013). Patients in the non-DEXA group showed higher BMI values compared to DEXA patients (31.27 versus 28.63, *p* = 0.018). The Charlson-Comorbidity Index (CCI) was significantly higher in the DEXA group compared to the non-DEXA patients [3.37 versus 2.42 for non-DEXA and DEXA, *p* = 0.013]. Average SAPS II and non-GCS SOFA scores were not different between groups [*p* = 0.854 and *p* = 0.963, respectively]. Interestingly, non-DEXA patients showed significantly worse hypoxemia at ICU admission compared to DEXA patients according to the Horowitz Index [139.35 versus 184.77; *p* = 0.001]. Need for intubation was the most frequent complication in DEXA and non-DEXA patients [77.4% versus 66.9%, *p* = 0.122], followed by bacteraemia [53.22% versus 57.98%, *p* = 0.494], kidney replacement therapy, and ECMO [43.54% versus 39.64% and 30.6% versus 31.4%, respectively]. Only 4/169 (6.5%) in the DEXA-group and 13/62 (7.7%) in the non-DEXA group were diagnosed with probable CAPA (*p* = 0.749). Time spent in the ICU was not different between the groups (14.16 days versus 13.68 days, *p* = 0.795). In total, 97/169 (57.40%) of DEXA and 37/62 of Non-DEXA patients (59.68%) survived their ICU stay (*p* = 0.756).

Incidence of probable COVID-19 associated pulmonary aspergillosis (pCAPA) in our cohort was 7.36% (17/231). Most patients showed multiple positive microbiology results that led to pCAPA diagnosis (10× GM > 1.0 ODI in BALF, 8× *Aspergillus* spp. culture in BALF, 4× *Aspergillus* spp. by PCR in BALF, 8× GM > 0.5 ODI in Serum). Patients with pCAPA and without CAPA (Non-CAPA) were similar in terms of demographics (Table 2). Average SAPS II score differed between groups (41.18 for pCAPA and 33.19 for non-CAPA, *p* = 0.036). There were significantly more complications in patients with pCAPA compared to non-CAPA patients, specifically, need for intubation and ECMO (100% versus 67.29%, *p* = 0.005 and 52.94% versus 29.44%, *p* = 0.044, respectively). This was also true for bacteraemia and kidney replacement therapy (88.24% versus 52.80%, *p* = 0.018 and 70.59% versus 38.32%, *p* = 0.009, respectively). Correspondingly, pCAPA patients spent more time in the ICU (22.71 versus 13.10 days, *p* = 0.002) and showed worse outcomes compared to the non-CAPA group (29.41% survivors versus 60.28%, *p* = 0.013). 

To gather risk factors for pCAPA development, a further logistic regression analysis was performed (Table 3). Within the cohort of patients presented here, the use of dexamethasone, disease severity, or the need for organ support were not associated with CAPA development.

A logistic regression analysis was performed to find independent risk factors for poor outcome as defined by death during ICU stay (Table 4). In this analysis, pCAPA was not an independent risk factor for death (*p* = 0.361). A higher burden of disease (higher SAPS II [*p* = 0.046], higher non-GCS SOFA [*p* = 0.004], lower Horowitz index [*p* = 0.006]), and an increased need for organ support (kidney replacement therapy [*p* = 0.03], ECMO therapy [*p* = 0.001]) were independently associated with a worse outcome. 

## 5. Discussion

In this retrospective analysis of critically ill patients with COVID-19, dexamethasone was not identified as a risk factor for a worse outcome or the development of probable CAPA (pCAPA). However, patients with pCAPA showed a higher burden of comorbidities, complications, and decreased survival compared to non-CAPA patients in a univariable analysis; pCAPA was not identified as an independent risk factor for death during ICU stay in a multivariable analysis even after adjusting for confounding factors.

Addressing this special group of patients is important since the incidence of CAPA ranges from 10% to 26% [7,8,9,10] with mortality rates up to 71% depending on CAPA definitions and study centre [7,8,9,10,11,12,13]. The incidence of CAPA in our study was as low as 7.36%. Mortality in the overall population was 42%, and in the group of pCAPA patients it was 70.59%. 

To the best of our knowledge, there are three meta-analyses that discuss the incidence and mortality of CAPA with incidence rates between 10% and 14.9% and mortality rates between 42.6% and 59.2% [8,9,10]. These studies are hardly comparable to our results. First, included studies not only used different CAPA definitions but also different Galactomannan thresholds. Second, included studies were published before and after the RECOVERY trial [16] was released. This makes an interpretation of the influence of dexamethasone on CAPA difficult. Third, two of three meta-analyses lacked information about disease severity of included patients. In Chong et al. [9], patients with CAPA had a higher severity of illness based on sequential organ failure assessment (SOFA) score (mean 9.37 vs. 7.27; MD 2.57; 95% CI 1.46–3.68; I^2^ = 0%; *p* < 0.001). In a study by Lahmer et al. [32], higher APACHE II scores were also independently associated with CAPA. 

ECMO is a last resort therapy in patients with severe ARDS. In a large multicentre study conducted in Portugal and Spain [33], survival of patients with COVID-19 on ECMO therapy was 56.4% during the first and second wave. The influence of ECMO therapy on the risk of IPA is not clear. Fekkar et al. [34] included a high number of COVID-19 patients with ECMO support (73/135; 54%), but this was not associated with invasive fungal infections or mortality. In a large series of 134 ECMO patients before the SARS-CoV-2 pandemic [35], IPA, but not *Aspergillus* colonisation, was associated with high long-term mortality. In our cohort, 31.16% (72/231) patients were treated with ECMO, whereas significantly more patients in the pCAPA group were put on ECMO therapy compared to non-CAPA patients (52.94% versus 29.44%). The need for ECMO support was also an independent risk factor for mortality in our study. 

Radiologic signs of CAPA are often difficult to distinguish from lesions caused by SARS-CoV-2. 2020 ECMM/ISHAM criteria [6] only require the presence of pulmonary or cavitating infiltrates on a chest CT for possible or probable CAPA diagnosis. EORTC/MSGERC definitions [36] on the other hand need specific *Aspergillus* lesions like air crescent sign or halo sign. In a study by Kariyawasam et al. [10], only 20% of CAPA patients showed an *Aspergillus*-specific radiological sign. Ghazanfari et al. [37] found no significant difference in the radiological appearance in COVID-19 patients with or without invasive fungal infections. These low numbers of classical fungal appearances on CT scans might be due to most patients with COVID-19 being immunocompetent. This is in line with recent recommendations [13] not to use CT scans of the chest to rule pulmonary fungal infections in patients with COVID-19 in or out.

Studies that also used the ECMM/ISHAM criteria published in 2020 [6] showed comparable results to our study. Prattes et al. [11] showed a CAPA incidence of 10.7% with significantly different mortality rates of 71% in CAPA and 43% in non-CAPA patients (*p* < 0.001). Results from a European medical mycology register [12] revealed an incidence of 13.5% and significantly different mortality rates of 52% (CAPA) and 39% (non-CAPA). 

The impact of corticosteroid use on the development of CAPA remains controversial. There are studies that highlighted an association between severe COVID-19 infections and CAPA with corticosteroids only [7] or combined with Interleukin-6 antagonists [21]. Others failed to confirm this association. Only one meta-analysis [9] studied the influence of corticosteroids with no effect on the development of CAPA. In a study by Ergün et al. [22], CAPA patients received corticosteroids more often, but this was not statistically significant in the univariable analysis. Among different kinds of corticosteroids during admission, only hydrocortisone use was significant in patients with CAPA compared to controls (38.5% vs. 12%, respectively, *p* < 0.001) in a study by Permpalung et al. [38]. This effect was not shown for dexamethasone. Since hydrocortisone is usually given to patients in septic shock, these results might only mirror the severity of disease in these patients. Three facts may explain this heterogeneity between studies. First, studies included patients before and after the release of the RECOVERY trial [16]. Second, fungal workup and CAPA definition varied between studies. Third, cumulative dosages of corticosteroids and substances differed substantially in the mentioned trials.

Why patients with severe course of COVID-19 are prone for infections caused by *Aspergillus* spp. is an ongoing matter of research. Patients with severe COVID-19 often have predisposing conditions that lead to poor immune responses, in particular type I or type III interferons (IFN) [39]. IFN are also essential for recruitment of neutrophil granulocytes (NG), one of the key player in defence of *Aspergillus* spp. [40]. In a study by Yang et al. [41], SARS-CoV-2 infection subverted IFN signalling in monocyte-derived dendritic cells through antagonizing STAT-1 phosphorylation. Dendritic cells also play a major role in orchestrating anti-fungal defence mechanisms by cytokine production. Severe immunopathology is a key feature of COVID-19 which is driven by uncontrolled inflammation [39]. This is confirmed by clinical observation where immunopathology is the dominating factor leading to a severe course of COVID-19.

This study has several limitations: first, data were collected retrospectively from one centre only. Second, only patients from the first and second pandemic waves were included in the study. With omicron being the dominant variant during the third wave, these results might have varied. Third, our intensive care unit is specialized in ARDS therapy, where the most severe cases of COVID-19 are treated. This could also bias the results. 

## 6. Conclusions

In this retrospective, single centre study, CAPA was not identified as an independent risk factor for mortality. A higher burden of disease, expressed by higher SAPS and non-GCS SOFA scores together with a lower Horowitz Index, as well as the need for organ support (ECMO and kidney replacement therapy) were significantly associated with a worse outcome. COVID-19 treatment with dexamethasone did not show any influence on mortality or risk for CAPA. Adequately treated CAPA was not an independent risk factor for mortality. Accordingly, CAPA might reflect patients’ severe disease instead of directly influencing outcome.

## Figures and Tables

**Figure 1 diagnostics-12-03049-f001:**
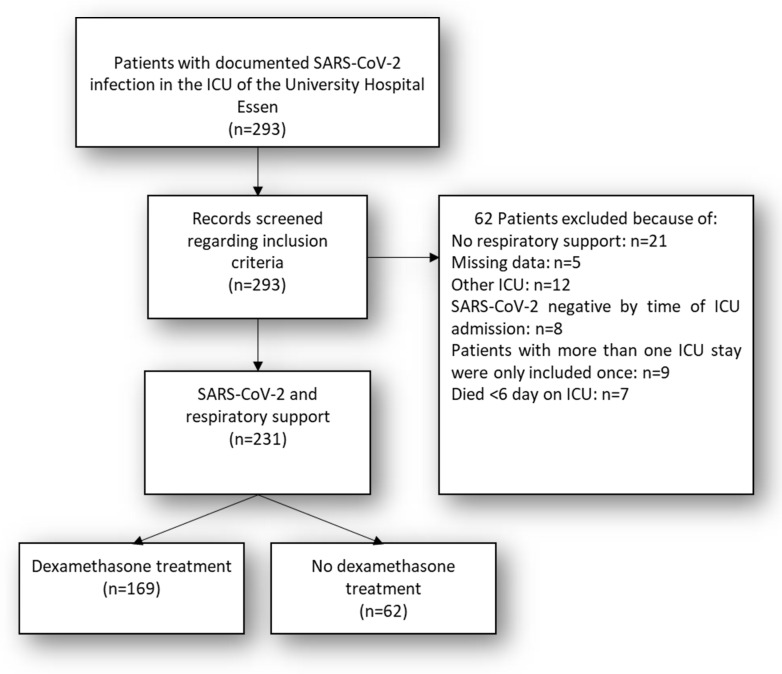
Study flow chart. n = 8 (SARS-CoV-2 negative by the time of ICU admission); n = 9 (Patients with more than one ICU stay were only included once).

**Table 1 diagnostics-12-03049-t001:** Baseline characteristics of DEXA- and Non-DEXA patients.

	DEXA (n = 169)	Non-DEXA (n = 62)	*p*
Demographics			
Age [years] ^#^	61.03 ± 13.82	57.11 ± 14.89	0.072
Male	117 (69.2%)	53 (85.5%)	0.013 *
Co-morbidity			
BMI [kg/m^2^] ^#^	28.63 ± 5.58	31.27 ± 7.82	0.018 *
Smoker	5 (8.1%)	11 (6.5%)	0.680
CCI ^#^	3.37 ± 3.11	2.43 ± 2.30	0.013 *
SAPS II ^#^	34.10 ± 15.14	33.67 ± 14.59	0.854
Horowitz Index ^#^	184.77 ± 131.79	139.35 ± 73.27	0.001 **
Non-GCS SOFA	7.65 ± 3.08	7.62 ± 3.25	0.963
Complications			
Bacteraemia	33 (53.2%)	98 (58.0%)	0.494
Candidemia	6 (9.7%)	10 (5.9%)	0.319
Probable CAPA	4 (6.5%)	13 (7.7%)	0.749
Invasive ventilation	48 (77.4%)	113 (66.9%)	0.122
Kidney replacement therapy	27 (43.5%)	67 (39.6%)	0.593
ECMO	19 (30.6%)	53 (31.4%)	0.917
Outcome			
Length of ICU stay [days] ^#^	14.16 ± 12.60	13.68 ± 12.41	0.795
Survivor	97 (57.4%)	37 (59.7%)	0.756

Data are presented as n (%). ^#^ Values are presented as mean ± standard deviation. * *p* < 0.05. ** *p* < 0.01. Abbreviations: *BMI* (Body Mass Index), *CCI* (Charlson Comorbidity Index), *SAPS II* (Simplified Acute Physiology Score II), *SOFA* (Sequential Organ Failure Assessment), *CAPA* (COVID-19 Associated Pulmonary Aspergillosis), *ECMO* (Extracorporeal Membrane Oxygenation), *ICU* (Intensive Care Unit).

**Table 2 diagnostics-12-03049-t002:** Baseline characteristics of pCAPA and non-CAPA patients.

	pCAPA (n = 17)	Non-CAPA (n = 214)	*p*
Demographics			
Age [years] ^#^	59.41 ± 14.38	58.06 ± 14.73	0.716
Male	14 (82.4%)	156 (72.9%)	0.395
Co-morbidity			
BMI [kg/m^2^] ^#^	31.45 ± 7.05	30.49 ± 7.41	0.608
Smoker	2 (11.76%)	14 (6.54%)	0.414
Leucocytes [10^6^/nl]	14.83 ± 7.34	24.24 ± 145.78	0.790
Dexamethasone	13 (76.5%)	156 (72.9%)	0.100
CCI ^#^	2.82 ± 2.24	2.67 ± 2.60	0.811
SAPS II ^#^	41.18 ± 9.05	33.19 ± 14.91	0.036
Horowitz Index ^#^	141.60 ± 71.38	152.36 ± 96.26	0.653
Non-GCS SOFA	8.59 ± 2.58	7.52 ± 3.25	0.192
Complications			
Bacteraemia	15 (88.24%)	113 (52.80%)	0.018 *
Candidemia	2 (11.76%)	14 (6.54%)	0.414
Intubation	17 (100%)	144 (67.29%)	0.005 **
Kidney replacement therapy	12 (70.59%)	82 (38.32%)	0.009 **
ECMO	9 (52.94%)	63 (29.44%)	0.044 *
Wave 1 ^a^	3 (7.69%)	36 (92.31%)	0.930
Wave 2 ^b^	14 (7.29%)	178 (92.71%)	0.930
Outcome			
Length of ICU stay [days] ^#^	22.71 ± 12.34	13.10 ± 12.19	0.002 **
Survivor	5 (29.41%)	129 (60.28%)	0.013 *

Data are presented as n (%). ^#^ Values are presented as mean ± standard deviation. * *p* < 0.05. ** *p* < 0.01. ^a^ n = 39. ^b^ n = 192. Abbreviations: *BMI* (Body Mass Index), *CCI* (Charlson Co-morbidity Index), *SAPS II* (Simplified Acute Physiology Score II), *SOFA* (Sequential Organ Failure Assessment), *CAPA* (COVID-19 Associated Pulmonary Aspergillosis), *ECMO* (Extracorporeal Membrane Oxygenation), *ICU* (Intensive Care Unit).

**Table 3 diagnostics-12-03049-t003:** Risk factors for pCAPA during ICU stay.

N = 155 ^§^	Odds Ratio (95% CI)	*p*
Multivariable Analysis		
Age	1.015 (0.957–1.076)	0.628
BMI [kg/m^2^]	1.024 (0.956–1.098)	0.497
Smoker	0.218 (0.307–1.543)	0.127
CCI	0.032 (0.767–1.389)	0.835
SAPS II	1.055 (0.971–1.147)	0.203
Non-GCS SOFA	1.023 (0.813–1.287)	0.847
Dexamethasone	0.585 (0.503–6.801)	0.669
Horowitz Index	1.003 (0.995–1.010)	0.463
Dialysis	1.146 (0.262–5.010)	0.857
ECMO	0.847 (0.255–2.811)	0.787
Time in ICU [days]	1.048 (1.004–1.095)	0.033 **
Waves	0.730 (0.044–12.101)	0.827

Abbreviations: *BMI* (Body Mass Index), *CCI* (Charlson Co-morbidity Index), *SAPS II* (Simplified Acute Physiology Score II), SOFA (Sequential Organ Failure Assessment), *CAPA* (COVID-19 Associated Pulmonary Aspergillosis), *ECMO* Extracorporeal Membrane Oxygenation), *ICU* (Intensive Care Unit). ^§^ Missing data (n = 76). ** *p* < 0.01.

**Table 4 diagnostics-12-03049-t004:** Risk factors for death during ICU stay.

N = 155 ^§^	Odds Ratio (95% CI)	*p*
Multivariable Analysis		
Age	1.038 (0.978–1.103)	0.218
BMI [kg/m^2^]	1.020 (0.947–1.097)	0.605
Smoker	1.499 (0.163–13.786)	0.721
CCI	1.0933 (0.845–1.414)	0.497
SAPS II	1.086 (1.001–1.177)	0.046 *
Non-GCS SOFA	1.423 (1.19–1.809)	0.004 **
Dexamethasone	0.722 (0.067–7.830)	0.789
probable CAPA	0.416 (0.063–2.731)	0.361
Horowitz Index	0.989 (0.982–0.997)	0.006 **
Kidney replacement therapy	0.246 (0.069–0.876)	0.030 *
ECMO	0.122 (0.036–0.414)	0.001 **
Length of ICU stay [days]	0.945 (0.902–0.990)	0.017 *
Waves	0.898 (0.066–12.282	0.936

Abbreviations: *BMI* (Body Mass Index), *CCI* (Charlson Co-morbidity Index), *SAPS II* (Simplified Acute Physiology Score II), *SOFA* (Sequential Organ Failure Assessment), *CAPA* (COVID-19 Associated Pulmonary Aspergillosis), *ECMO* (Extracorporeal Membrane Oxygenation), *ICU* (Intensive Care Unit). ^§^ Missing data (n = 76). * *p* < 0.05. ** *p* < 0.01.

## Data Availability

Data available on request due to restrictions of privacy or ethical. The data presented in this study are available on request from the corresponding author.

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
