# Peer review of "Effect of Dexamethasone on the Incidence and Outcome of COVID-19 Associated Pulmonary Aspergillosis (CAPA) in Critically Ill Patients during First- and Second Pandemic Wave—A Single Center Experience"

_diagnostics, 2022, doi:10.3390/diagnostics12123049_

Round 1
Reviewer 1 Report
In this paper, the authors investigated efect of dexamethasone on the incidence and outcome of COVID-19 associated pulmonary aspergillosis (CAPA) in critically ill patients. This topic is clinically important considering the difficulties in treating severe and critical patients with COVID-19. The results provided new reference for the treatment and management of comorbidity of COVID-19. However, some improvements are essential in introduction, methods, and discussion to meet the publishable standards.
1. Introduction: Please enrich the introduction on COVID-19 severity. The severity of COVID-19 depends on many physiological factors of which some are explainable in AI-assisted severity classification, whereas the pathological mechanism of others are still unclear (Refer: 10.3390/ijerph191710665).
2. Methods: “A t-test was used with a p < 0.05 suggesting significant differences.” This sentence is not in accordance with the next one, and can be removed. Please also provide the detailed predictions for different statistical tests, e.g., how did you define that the normal distribution was satisfied? In what condition the Fisher’s exact test was used instead of Chi-squared test?
3. Discussion: If possible, please provide some pathological explanation on the results.
4. Minor problems: There are many minor error in spelling, grammar, and format, e.g., semicolons are misused in the abstract, line 185: ” Chi-quadrat”, etc. These errors largely affected the quality of this manuscript. I advise the authors to find a native English speaker to proofread the manuscript.
Author Response
Thank you very much for your suggestion. Attached is the reply.

Reviewer 2 Report
In this manuscript, the authors describe the potential risks of administering dexamethasone in COVID-19 patients developing CAPA. The text is easy to read and it is well written, congratulations. However, certain aspects need to be solved before publication, in my opinion, although minor in general:
- Since data are from initial stages of the COVID-19 pandemic, I would suggest to clearly state it, including in the title. Currently, with omicron, these results might have varied.
- Please use the terms univariaBLE and multivariaBLE
- Line 41: Has this % changed over the pandemic?
- Please, mention genera and species in italics
- Line 53: I would suggest to include Salmanton-García et al. Emerging Infectious Diseases 2021, as several rates from different centres and geographical settings are provided. The publications mentioned for these rates are single centre experiences
- Line 54: Corticosteroids might not be the preferred drug for COVID-19 patients anymore. Thus, I think it is of relevance to mention the "historical" background of your data
- Line 100: If the standard of case in your institution has changed, please specify the latest time limit
- Lines 183 and 187 are redundant
- Line 228: How did you define "poor outcome"? If mortality, why do you perform a logistic regression instead of a Cox regression?
- Please, improve a bit the image quality of figure 1
- In figure 1 dexa group is 171, and in table 1 169. This mismatch with numbers applies also to non-dexa
- If table 3 is related to mortality and table 4 to development of CAPA, I would suggest to follow a chronology and make CAPA development before death
- Please, include the meaning of the variables the very first time you mention them (pCAPA is explained in discussion, but you also use it in results)
- Line 321: Why don´t you use the same references for this topic as in introduction?
- Line 327: Why don´t you compare your rates to studies using the same definition, like Prattes et al, for instance?
- I miss a paragraph regarding limitations
- Table S1 makes quite clear that the data are "old". Thus I suggest to mention this more clear, even in the title, like "Data from 1st and 2nd pandemic wave" or similar
Author Response

(The authors gave the same response as above.)

Round 2
Reviewer 1 Report
Thanks for the update. The majority of my earlier comments have been well addressed. There are some minor errors (e.g., missing punctuations) that need further improvement. I encourage the authors to find a native English speaker for proofreading.